# Deletion of the *Aspergillus niger* Pro-Protein Processing Protease Gene *kexB* Results in a pH-Dependent Morphological Transition during Submerged Cultivations and Increases Cell Wall Chitin Content

**DOI:** 10.3390/microorganisms8121918

**Published:** 2020-12-02

**Authors:** Tim M. van Leeuwe, Mark Arentshorst, Gabriel Forn-Cuní, Nicholas Geoffrion, Adrian Tsang, Frank Delvigne, Annemarie H. Meijer, Arthur F. J. Ram, Peter J. Punt

**Affiliations:** 1Institute of Biology Leiden, Microbial Sciences, Leiden University, Sylviusweg 72, 2333 BE Leiden, The Netherlands; t.m.van.leeuwe@biology.leidenuniv.nl (T.M.v.L.); m.arentshorst@biology.leidenuniv.nl (M.A.); peter.punt@ddna-biotech.com (P.J.P.); 2Institute of Biology Leiden, Animal Sciences, Leiden University, Einsteinweg 55, 2333 CC Leiden, The Netherlands; g.forn-cuni@biology.leidenuniv.nl (G.F.-C.); a.h.meijer@biology.leidenuniv.nl (A.H.M.); 3Centre for Structural and Functional Genomics, Concordia University, Montreal, QC H4B1R6, Canada; nicholas.geoffrion@concordia.ca (N.G.); adrian.tsang@concordia.ca (A.T.); 4TERRA Teaching and Research Centre, Gembloux Agro-Bio Tech, University of Liège, Avenue de la Faculté, 2B, 5030 Gembloux, Belgium; f.delvigne@uliege.be; 5Dutch DNA Biotech, Hugo R Kruytgebouw 4-Noord, Padualaan 8, 3584 CH Utrecht, The Netherlands

**Keywords:** cell wall, chitin, morphology, RNA-seq, batch-cultivation, biofilm formation

## Abstract

There is a growing interest in the use of post-fermentation mycelial waste to obtain cell wall chitin as an added-value product. In the pursuit to identify suitable production strains that can be used for post-fermentation cell wall harvesting, we turned to an *Aspergillus niger* strain in which the *kexB* gene was deleted. Previous work has shown that the deletion of *kexB* causes hyper-branching and thicker cell walls, traits that may be beneficial for the reduction in fermentation viscosity and lysis. Hyper-branching of *∆kexB* was previously found to be pH-dependent on solid medium at pH 6.0, but was absent at pH 5.0. This phenotype was reported to be less pronounced during submerged growth. Here, we show a series of controlled batch cultivations at a pH range of 5, 5.5, and 6 to examine the pellet phenotype of *ΔkexB* in liquid medium. Morphological analysis showed that *ΔkexB* formed wild type-like pellets at pH 5.0, whereas the hyper-branching *ΔkexB* phenotype was found at pH 6.0. The transition of phenotypic plasticity was found in cultivations at pH 5.5, seen as an intermediate phenotype. Analyzing the cell walls of *ΔkexB* from these controlled pH-conditions showed an increase in chitin content compared to the wild type across all three pH values. Surprisingly, the increase in chitin content was found to be irrespective of the hyper-branching morphology. Evidence for alterations in cell wall make-up are corroborated by transcriptional analysis that showed a significant cell wall stress response in addition to the upregulation of genes encoding other unrelated cell wall biosynthetic genes.

## 1. Introduction

Filamentous fungi are industrially used to produce a range of products, from organic acids, antibiotics and other metabolites to enzymes and (heterologous) proteins. However, industrial-scale fermentation using filamentous fungi is typically limited by limited oxygen supply due to the high viscosity of the fermentation broth at high mycelial growth densities. These conditions impair homogeneous mixing, are very energy-demanding, and cause stress to the fungus due to the high amounts of hyphal shearing. Fluctuations in nutrient and oxygen levels caused by sub-optimal mixing within the large-scale fermentation vessels can subsequently lead to additional metabolic stress and shifts, diminishing batch-to-batch consistency and product quality [1,2].

In addition to high-viscosity problems, intensive use of filamentous fungi as cell factories at an industrial scale produces large amounts of spent mycelium left over as a by-product. Spent mycelium has been suggested to be a source of chitin and chitosan products that can be extracted from the fungal cell wall to be used for many applications in medicine and agriculture [3,4,5]. To both outcompete the current supply of chitin from crustacean shell waste and to make chitin yields a profitable option, the optimization of extraction and high chitin levels is required [3,6]. Efforts have been made to do so by genetic modification of the chitin biosynthetic pathway or through alterations in fermentation conditions [7,8,9]. Additionally, we recently reported on the identification of two *Aspergillus niger* UV-mutants that showed increased cell wall chitin [10,11]. These efforts contribute to exploring the use of spent mycelium as an added-value product rather than waste output.

In the endeavor to address both the issue of fermentation (mycelial) viscosity and increase chitin production, we turned to an *A. niger kexB* deletion strain that is already known for impaired pro-protein processing. The deletion of *kexB* has already been shown to be beneficial for the secretion of fusion proteins consisting of a well secreted carrier and proteolytically sensitive proteins [12]. In *A. niger*, *kexB* (also named *pclA* in the literature) was shown to be implicated in the processing of dibasic cleavage sites of secretory proteins [12,13]. The *A. niger* KexB protein is the homologue of *Saccharomyces cerevisiae* Kex2p, a Ca^2+^-dependent serine protease that is responsible for processing dibasic Lys-Arg or Arg-Arg cleavage sites for the maturation of secreted proteins [14]. Designated proteins that pass through the secretory pathway are processed in the Golgi apparatus where kexin proteins reside due to their Golgi-retention signal [15,16]. These Kex enzymes are important in ascomycete alpha-pheromone processing [17,18], first discovered in yeast, where two to four copies are the alpha-pheromone, processed into peptides by *kex2* [19,20,21]. Additionally, Kex2 was shown to be important in subsequent steps of mating during cell fusion [22]. Furthermore, in filamentous fungi, KexB is implicated in the processing of cyclic and modified peptides [23]. Previous reports have also shown that the *A. niger ∆kexB* strain displayed shorter, visibly thicker hyphae and a hyper-branching morphology [12,13,24]. Shorter hyphae and smaller pellets are ideal traits to reduce stirring viscosity in fermenter conditions, but the actual performance of this strain under fermentation conditions has not been fully explored. Cell wall compositional assessments have never been performed in any *kexB* mutant strain in *A. niger*. Besides the above-mentioned processing targets of yeast and fungal kexins, many more putatively kexin-processed proteins can be found in fungal proteomes and inferred from the corresponding genomes.

A knockout of *KEX2* in *Candida albicans* resulted in reduced virulence with 147 predicted proteins that were identified as potential Kex2p targets that relate to cell wall construction and modification, including hydrolases, adhesins, cell wall components, and outer membrane proteins [25]. Previous work in *A. oryzae* has also shown that the deletion of *kexB* affects cell wall synthesis and activation of the CWI pathway by MAPK phosphorylation assays [24,26,27]. Using prediction algorithms such as v.1.0b ProPeptide Cleavage Site Prediction (ProP), a plethora of putative KexB targets can be identified, which are only indicative for potential targets and lack biological interpretation of any relation to the observed pleiotropic effects, such as shorter hyphae and a hyperbranching phenotype. Interestingly, this growth phenotype has clearly been shown on plates at pH 6.0, but not at pH 5.0, suggesting a pH-dependent phenotype [28], but a detailed study how the *ΔkexB* strain behaves during submerged growth is lacking. Here, we investigated the role of KexB on hyphal morphology in pH-controlled batch cultivations. In doing so, we aimed to investigate the impact of deleting *kexB* in *A. niger* on the cell wall composition with respect to its shorter and thicker hyphae.

## 2. Materials and Methods

### 2.1. Strains, Media, Growth Conditions

The strains used in this study can be found in Table 1. All media were prepared as described [29]. In all cases, minimal medium (MM) contained 1% (*w*/*v*) glucose, 1.5% agar (Scharlau, Barcelona, Spain) and was not supplemented unless otherwise specified. Complete medium (CM) contained 0.1% (*w*/*v*) casamino acids and 0.5% (*w*/*v*) yeast extract in addition to MM. Strains were inoculated from −80 °C glycerol stocks onto fresh CM plates and were allowed to grow and sporulate for 5–7 days at 30 °C, prior to spore harvesting. Spores were harvested by the addition of 15 mL of 0.9% (*w*/*v*) NaCl to CM spore plates and were carefully scraped from the surface with a cotton swab. In case of harvesting spore plates for bioreactor cultivations, 0.05% Tween-80 was added to a 0.9% (*w*/*v*) NaCl solution to prevent spore clumping. Spore solutions were poured over sterile cotton filters (Amplitude™ Ecocloth™ Wipes, Contec Inc., Spartanburg, SC, USA) to remove large mycelial debris. Spore solutions were counted using Bio-Rad TC20™ Automated Cell Counter (Bio-Rad Laboratories, Inc. Hercules, CA, USA) using Counting Slides, Dual Chamber for Cell Counter (Cat#145-0011, Bio-Rad Laboratories, Inc. Hercules, CA, USA).

### 2.2. Bioreactor Cultivation

Controlled batch cultivations for *A. niger* strains N402 and the *ΔkexB* strain were performed in 6.6 L BioFlo bioreactors (New Brunswick Scientific, Edison, NJ, USA), as previously described [32]. A batch of 21 L MM containing 0.75% D-glucose was made by adding 1 L filter-sterilized (0.2 µm pore) glucose (15.75% *w*/*v*) solution to a freshly autoclaved volume of 20 L MM (no carbon source) as described above. Allowing 1 day of dissolving and a check for contamination, 5 L of MM 0.75% glucose was added to each bioreactor directly after autoclaving. Temperature, acidity and stir speed were set to and kept at 30 °C, pH 3 and 250 rpm, respectively. The pH was controlled by addition of titrants (2 M NaOH and 1 M HCl). Sparger aeration of 1 L/min was left on to allow oxygen saturation of the medium prior to inoculation. Next, aeration was set to headspace only and 1.5 mL 10% *w*/*v* Yeast Extract was added to the medium to promote homogeneous germination for the to-be-added spores. Subsequently, a total of 5 × 10^9^ (10^6^ sp/mL) spores were added to the medium using a concentrated spore solution. A germination time of approximately 4–5 h was maintained, preceding the addition of polypropylene glycol P2000 anti-foam agent, increasing agitation to 750 rpm and changing aeration from headspace to sparger only (1 L/min). Oxygen, base and acid consumption were monitored, and samples were taken at regular intervals to obtain biomass, culture filtrate and microscopy samples. Biomass was harvested by applying a vacuum over Whatman™ Glass Microfiber Filter (GF/C™) (diameter 47 mm, CAT No.1822-047, Buckinghamshire, UK). Samples were all quickly frozen in liquid nitrogen prior to storage at −80 °C. Biomass accumulation through time was gravimetrically determined by lyophilizing designated samples from the corresponding broth culture mass.

### 2.3. Biofilm Cultivations

Biofilm cultivations were performed in a 4 stirred-tank mini-bioreactors platform (DASGIP DASbox Reactor SR0250ODLS, Eppendorf AG, Hamburg, Germany). For promoting biofilm formation, the stirring device was completely removed in order to leave space for two sheets of stainless-steel 316 L wire gauze. These metal sheets were used as a support promoting biofilm growth. Each bioreactor was filled with 200 mL of MM and operated at 30 °C. The pH level was maintained at 5 or 6 depending on the experiment (controlled by the addition of NH_4_OH or H_3_PO_4_) and the air flow rate was adjusted to 200 mL/min. Dissolved oxygen was measured using PSt1 optical sensors linked to an OXY-4 oxygen meter (Presens Precision Sensing, Regensburg, Germany). Each bioreactor was initially inoculated with spores in order to reach 10^6^ sp/mL. At the end of the cultivation, metal sheets were removed from the bioreactor for estimating the biofilm wet (in this case, the sheets were left for 30 min in a beaker for removing excess of liquid before mass measurement) and dry (estimated after keeping the sheets at 105 °C for 24 h) weight.

### 2.4. Microscopy 

Pellet morphology samples were taken at 100% biomass and visualized in a Zeiss Observer confocal laser-scanning microscope (Zeiss, Jena, Germany). Images were processed and analyzed using FIJI (ImageJ) software [33].

### 2.5. Cell Wall Isolation and Chitin Analysis

Cell wall samples were isolated as previously described [11]. In short, dried mycelium was frozen in liquid N_2_ and were ground to break open the cells. Samples were washed to remove intracellular debris and proteins, three times with 1 M NaCl and three times with MilliQ ultrapure water (MQ). Supernatant was carefully discarded prior to the next washing step. Cell wall samples were lyophilized after washing steps for 48 h. Cell wall isolation, hydrolysis and chitin content analysis, measured as total glucosamine, were performed as described previously [11]. Cell wall glucosamine measurements from independent replicate experiments are expressed as means ± SE.

### 2.6. RNA Isolation and RNA-Sequencing

RNA was isolated from mycelial biomass samples obtained from batch-cultivated *A. niger* strains N402 and *∆pclA* (*∆kexB*), using TRIzol (Invitrogen). RNA was purified afterwards with NucleoSpin RNA Clean-up kit (Macherey-Nagel) with DNase treatment. Concentration and quality of the RNA was determined using a NanoDrop 2000 spectrophotometer (Thermo Scientific, Breda, The Netherlands) and by gel-electrophoresis, respectively. RNA sample were sent to Centre d’expertise et de services Genome Québec for sequencing using the HiSeq4000 technology. Sequencing data are available under GEO accession number GSE151618.

### 2.7. Transcriptomic Analysis

Raw RNA-seq read sets were retrieved from the Centre d’expertise et de services Génome Québec’s Nanuq portal, and pre-processed with BBDuk from the BBTools package (https://sourceforge.net/projects/bbmap) to trim sequencing adapters and remove reads derived from PhiX and ribosomal RNA. The transcriptome of *A. niger* NRRL3 (v. 20140311) was retrieved from the *jgi* Genome portal [34], and the raw reads were mapped to the transcriptome using Salmon v0.14.1 [35]. The libraries were imported in RStudio 1.2.5001 (RStudio: Integrated Development for R. RStudio, Inc., Boston) running R 3.6.1 (R Development Core Team 3.6.1) using txtimport v.1.12.3 [36]. Differential gene expression was assessed via pairwise comparisons using DESeq2 v1.24.0 [37], using the design~mutation (Padj ≤ 0.05). Updated gene length data for the NRRL3 genome were retrieved from the *JGI* Genome portal. The full code is available at https://github.com/gabrifc/rnaseq_analysis_kexB.

### 2.8. Gene Knockout of kexB in TLF39

A deletion of *kexB* was introduced in the seven-fold *crh* knockout strain (TLF39) using a split marker approach [29]. TLF39 (Table 1) was transformed after protoplastation as described previously [29]. Using the split marker approach for single-gene knockout, the entire *kexB* ORFs was deleted by replacement with the hygromycin B selection marker [38]. Primer sequences are available upon request. Approximately 2 µg of DNA per flank was added to protoplasts for transformation. Transformation plates were incubated on MMS for 6 days at 30 °C. Transformed colonies were single-streaked on MM twice for purification and were genotyped using diagnostic PCR (data not shown). 

### 2.9. Cell Wall Sensitivity Assays

Cell-wall-disturbing compounds Calcofluor White (CFW), Congo Red (CR), Caspofungin (CA), sodium dodecyl sulfate (SDS, 0.004% and 0.005%), were added to MM plates. Spores were harvested as described above, counted, and serially diluted into 2000, 200, 20 and 2 spores/µL, and 5 µL of respective dilutions was spotted on MM plates containing cell-wall-disturbing compounds. Plates were incubated for 3–5 days at 30 °C.

## 3. Results 

### 3.1. Disruption of kexB Shows a pH-Dependent Phenotype between pH 5.0 and pH 6.0 in Fermenters

The deletion of *kexB* in *A. niger* is known to result in a pH-dependent morphological phenotype on solid medium [13]. On pH 6.0 buffered agar plates, the *ΔkexB* strain displays a hyper-branching and compact phenotype, whereas at pH 5.0 growth is like wild type [28]. To analyze the effect of the pH on the morphology during stirred submerged growth, we cultivated both the parental strain (N402) and *ΔkexB* at pH 5.0, 5.5 and 6.0. Batch cultivations were performed at pH-controlled conditions using 0.75% glucose as carbon source. Dry weights were used to determine both the maximal specific growth rate (µmax) and maximum biomass, which are listed in Table 2. At pH 5.0, both the growth rate (0.200 · h^−1^) and maximum biomass (4.15 g_DW_ · kg^−1^) of the *ΔkexB* strain were found to be similar to the wild type (0.187 · h^−1^, 3.66 g_DW_ · kg^−1^). The hyphal morphology of the *ΔkexB* strain shows a wild type-like phenotype at pH 5.0, although the *ΔkexB* strain showed a slightly more open pellet morphology (Figure 1). Submerged growth at pH 5.5 showed slightly shorter hyphae and increased branching in the case of the *ΔkexB* strain compared to wild type, as can be seen in Figure 1. The growth rate and maximum acquired biomass at pH 5.5 for the *ΔkexB* strain (0.194·h^−1^, 4.19 g_DW_·kg^−1^) was similar to wild type (0.175·h^−1^, 3.68 g_DW_·kg^−1^) and also similar to the conditions at pH 5.0. When grown at pH 6.0, the *ΔkexB* strain again showed a similar growth rate (0.211·h^−1^) and maximum biomass accumulation (4.31 g_DW_·kg^−1^) as at other pH conditions, but importantly, showed a very clear compact pellet phenotype (Figure 1). Cultivation of N402 at pH 6.0 resulted in severe biofilm formation on the walls of the fermenters. As a result, submerged biomass samples represented an incorrect measurement of both the submerged growth rate (0.063·h^−1^) and amount of in-broth maximum biomass (0.64 g_DW_·kg^−1^). Microscopic analysis of the N402 pellets at pH 6.0 showed a similar hyphal morphology compared to pH 5.0 and pH 5.5. In general, base consumption to maintain the desired culture pH in all cultures was largely similar, as was published earlier for N402 [39].

### 3.2. Disruption of kexB Displays Reduced Biofilm Formation and More Compact Biofilm Structure

As observed in the previous section, classical filamentous growth (i.e., without hyper-filamentation) displayed by wild type (N402) strain leads to severe biofilm proliferation on bioreactor walls. It thus seems that the hyper-filamentous phenotype exhibited by *∆kexB* strain did not promote biofilm formation. In order to challenge this hypothesis, both strains were cultivated in a biofilm reactor at two different pH values (i.e., 5 and 6). This cultivation mode involves a standard lab-scale bioreactor where the mechanical stirring device is removed and replaced by two sheets made of stainless-steel wire gauze (Figure 2A). This device has been evaluated for the cultivation of *Aspergillus oryzae* in a previous study and observations have pointed out that fungal growth occurs exclusively on the metal sheets in this cultivation device [40], making the growth process fully dependent on the adhesion capacity of the strains. Qualitatively, wild type strain (N402) at both pH levels and *∆kexB* strain at pH 5 displayed similar biofilm morphologies, i.e., mycelium with a relatively low ramification frequency. On the other hand, at pH 6, a totally different biofilm morphology was displayed, with this one being composed of several hyper-ramified mycelial clumps instead of a continuous layer of mycelium (see microscopy images at Figure 2A).

The two strains also exhibited differences at the quantitative level (Figure 2B), with biofilm dry weights for wild type strain (N402) cultivated at pH 5 and 6 of 2.95 and 2.39, respectively (mean values), whereas the mean dry weights of 1.74 and 1.31, respectively, were observed for the *∆kexB* strain. The pH-dependent hyper-filamentation phenotype of the *∆kexB* strain is thus also observed in the biofilm mode of cultivation, leading to reduced attachment to the metal supports and a much thicker biofilm structure with reduced water sorption capacity (Figure 2C). 

### 3.3. Cell Wall Chitin Content Is Increased in ΔkexB Irrespective of Morphology

To assess the relationship of different hyphal branching morphologies at pH 5.0, 5.5 and 6.0, and cell wall chitin content, we isolated cell walls from maximum biomass samples that were obtained from the bioreactor cultivations described above. Cell wall samples were hydrolyzed in triplicate and total glucosamine content was assessed using a colorimetric assay (Section 2.5). The results of glucosamine measurements are shown in Figure 3. During all cultivations, the *ΔkexB* strain displayed an increase in chitin content compared to the wild type. The increase in percentage of cell wall glucosamine for the *ΔkexB* strain compared to the wild type were found to be 35.0 ± 13.3%, 27.3 ± 5.51% and 29.6 ± 6.62% at pH 5.0, 5.5 and 6.0, respectively. Looking at absolute numbers of cell wall glucosamine per cell wall dry weight, we find that total glucosamine levels increase proportionally with the culture medium pH for both strains. Hence, cell wall glucosamine increases with increasing pH with about 20% between pH 5.0 and 6.0, and at all pH values the chitin content of the *ΔkexB* strain is about 30% higher compared the wild type. 

### 3.4. Genome-Wide Expression Profiling Reveals Changes in Expression of Cell Wall Biosynthetic Genes

To gain better insight to transcriptomic responses in relation to cell wall biosynthetic changes that occur as a result of lacking KexB, we decided to look into the transcriptome by performing RNA-sequencing on the RNA extracted from wild type and the *ΔkexB* mutant grown at pH 5.5. A pH of 5.5 resulted in the most comparable growth conditions for both the wild type and the *∆kexB* strain. To obtain biological duplicates, additional bioreactor cultivations at pH 5.5 were performed and resulted in similar growth rates and maximum biomass accumulation, as shown before for both wild type (0.167·h^−1^, 3.54 g_DW_·kg^−1^) and the *∆kexB* strain (0.180·h^−1^, 4.30 g_DW_·kg^−1^) (Table 2). RNA was isolated from culture samples in the exponential growth phase at 90% of the maximum attained biomass. Duplicate batch-culture cultivations for both wild type and mutant were used to obtain a total of four RNA-sequencing samples. Following DeSeq2 analysis (Section 2.7), we found 2461 transcripts (1163 up, 1298 down)—approximately 21% of the 11846 total transcripts—to be differentially expressed between the mutant strain and the wild type (adjusted *p*-value ≤ 0.05) (expression data in Appendix A). 

For this study, we focused our interest on investigating to what extent cell wall biosynthesis was affected by the deletion of *kexB*. To assess this, we checked for differential expression of all genes involved in 82 cell wall biosynthesis in *A. niger,* as reported by Pel et al., 2007 [41]. Table 3 summarizes all cell-wall-related genes that were differentially expressed in the *ΔkexB* strain compared to N402. Differential expression was found for 29 out of 82 cell-wall-related genes involved in synthesis of all major components of the cell well, including α-glucan, β-glucan and chitin, and the modification thereof (Table 3, Appendix A). The results for increased cell wall chitin (Figure 3) were corroborated by transcriptional upregulation of gene encoding the rate-limiting step in chitin precursor synthesis (UPD-*N*-acetylglucosamine), *gfaA* (FC 1.19) and *gfaB* (FC 6.22). Other genes involved in the synthesis of UPD-*N*-acetylglucosamine (*gnaA*, *pcmA* and *ugnA*) were not differentially expressed. Additionally, five chitin synthases were found to be upregulated in the *∆kexB* strain (*chsB*, *chsC*, *chsD*, *chsE* and *chsG*), whereas four out of seven putative chitin-to-glucan crosslinking enzymes were found to be differentially expressed (*crhA*, and *crhD* upregulated, and *crhB* and *crhF* downregulated) along with four differentially expressed chitinases (*cfcD* and NRRL3_09653 upregulated, and *cfcG* and NRRL3_04221 downregulated). 

Next to the upregulation of chitin-related transcripts, we observed the upregulation of *agsA* (FC 5.52), a gene known to be induced upon activation of the cell wall integrity (CWI) pathway in an RlmA-dependent manner [42]. We checked for upregulation of additional genes involved in the CWI response, as previously published [43,44]. Many of the reported genes were found to be represented in the differential dataset including *bxgA*, *gelA*, *chsE*, *crhD*, *chsB*, *dfgC* and *wscB* (highlighted by asterisks in Table 3). However, some cell-wall stress-related genes, such as chitin synthesis genes *gfaA* and *gfaB*, are not listed in this table. Taken together, the transcription data corroborate the observed increase in cell wall chitin and indicated that most of the CWI pathway is induced in a *ΔkexB* strain.

### 3.5. Cell Wall Integrity Is Affected by Disruption of kexB

To test whether the deletion of *kexB* affects the integrity of the cell wall, we exposed both the parental strain (N402) and *ΔkexB* to a series of cell-wall-disturbing compounds around pH 6.0 on plates—where the growth effect of *ΔkexB* is most prominent—that respond to changes in chitin content: Sodium dodecyl Sulfate (SDS), Caspofungin (CA), CalcoFluor White (CFW) and Congo Red (CR). As can be seen from Figure 4, the *ΔkexB* strain is more sensitive to SDS, CFW and CR than the wild type, while slight resistance towards CA was observed for the *ΔkexB* strain in this assay. 

In addition to an increase in chitin content for the *ΔkexB* strain, the transcriptional response showed differential expression for the majority of the chitin-to-glucan crosslinking *crh* family (Table 3). To investigate the importance of these enzymes in the construction and modification of existing chitin in the cell walls of the *ΔkexB* strain, we opted for knockout *kexB* in an existing seven-fold *crh* deletion [31]. We found that deleting *kexB* in the seven-fold *crh* deletion strain resulted sporulation deficiency. Despite the impact on sporulation, we neither observed a vegetative growth deficiency nor did we find a significant exacerbation of sensitivity towards any of the tested compounds in the *ΔcrhA-G* + *ΔkexB* double deletion mutant (Figure 4).

## 4. Discussion

In this study, we invoked the impact of deleting *kexB* in *A. niger* on hyphal morphology and cell wall composition by performing phenotypic, cell wall and transcriptomic analysis. We show that the characteristic hyper-branching pH-dependent phenotype of the *ΔkexB* strain, as reported on solid agar plates, is also prevalent in submerged solid support growth and submerged pH-controlled batch cultivations at pH 6.0. The phenotypic plasticity of the *ΔkexB* strain transitions between pH 5.0 and pH 6.0; pH-controlled conditions revealed that the *ΔkexB* strain morphology resembles the wild type at pH 5.0, shows slightly shorter hyphae and smaller pellets at pH 5.5, and shows severe hyper-branching and tiny pellets at pH 6.0 (Figure 1). Despite the differences in morphology, both the growth rate and maximum biomass of the *ΔkexB* strain were not impaired and were even slightly higher than the wild type across all pH conditions. It is interesting to note, however, that wild type growth at pH 6.0 resulted in severe biofilm formation at the fermenter walls, whereas the *ΔkexB* strain did not. 

To extend the observation that *kexB* may be involved in surface attachment (i.e., biofilm formation), we grew both wild type and *ΔkexB* in specific biofilm reactors. Under these very specific cultivation conditions, it was shown before that fungal growth occurs only on the metal sheets and not (or in very limited amount) in the liquid phase [40,45]. On these metal sheets, the *ΔkexB* strain showed reduced biofilm formation compared to the wild type at both pH 5.0 and pH 6.0. These observations showed that the deletion of *kexB* limits the proliferation of the fungal biomass on the metal support. Additionally, we found that the hyper-branching phenotype of the *ΔkexB* strain was only observed at pH 6.0 and is very similar to the batch cultivation conditions at pH 6.0. Additionally, the biofilm of the *ΔkexB* strain produced at pH 6.0 is of a totally different structure to the one observed for the wild type, as well as for the *ΔkexB* strain at pH 5.0. This mycelial layer of hyper-branching phenotype growing on the metal support is more compact and displayed reduced water sorption capacity. It is important to note that it remains unknown to what extent the morphology and biofilm composition will affect secretion and total production parameters, and these may be topics of future research. Despite these different mycelial properties of the hyper-branching morphology, a deletion of *kexB* reduced the ability to form biofilms at both pH conditions compared to the wild type. Furthermore, the increase in pH from 5.0 and 6.0 appeared to affect both wild type and *ΔkexB,* as a similar percentile drop in biofilm dry weight was observed between pH 5.0 and 6.0 (Figure 2B). Taken together, these data show that the *kexB* deletion shows a pH-dependent morphology at pH 6.0 without affecting growth rate or maximum biomass formation under submerged batch cultivation conditions. Additionally, the hyper-branching morphology showed less mycelial clumping and will reduce the stirring viscosity due to tinier pellets. As such, the deletion of *kexB* may provide a valuable industrial candidate strain for fermentation in near-neutral pH conditions. Additionally, we observed that the *kexB* deletion resulted in visibly thicker cell walls that could be used as an added-value post-fermentation product.

Next, we investigated the cell wall composition of the *∆kexB* strain and found an increase in cell wall chitin compared to the wild type; a trait that is often coupled with activation of the cell wall integrity to increase the strength of the cell wall [10,11,46,47,48,49,50], and increased susceptibility towards CFW and resistance towards CA [47,48,51]. The observed increase in chitin content from bioreactor cultivations was corroborated by the transcriptional upregulation of *gfaA*, and its paralogue *gfaB*, that are required for chitin precursor (UDP-*N*-acetylglucosamine) synthesis, known as the rate-limiting step for chitin synthesis [46,52]. In addition, upregulation of multiple chitin synthases and downregulation of chitinases *cfcD*, *cfcG* and the N402 lineage-specific chitinase NRRL3_04221 were found. This transcriptional response is similar to that of the UDP-galactopyranose mutase A (*ugmA*) cell wall mutant, lacking cell wall galactofuranose, that also increased cell wall chitin and chitin modifications by upregulation of *gfaA* and *gfaB*, *chsB* and *chsE* and severe downregulation of chitinase *ctcA* [44,53]. 

Aside from the increased chitin synthesis in the *ΔkexB* strain, chitin-modifying enzymes also appear to play a role in the response to KexB-deficiency, as can be inferred from the upregulation of *crhA* and *crhD*, and downregulation of *crhB* and *crhF* (Table 3). However, the combined deletion of all *crh* genes and *kexB* did not result in increased susceptibility to cell-wall-disturbing compounds. We did observe reduced sporulation, an effect that was also observed when the deletions of the seven-fold *crh* family and *ugmA* were combined [31]. In case of the *ugmA* strain, it has clearly been described that the CWI pathway is activated. Here, the *ΔkexB* strain also showed a large number of differentially expressed cell wall biosynthetic genes that are known to be upregulated during the CWI response. Moreover, the *kexB* deletion has been reported to induce the CWI response pathway by means of MAPK phosphorylation in *A. oryzae* and *A. fumigatus* [24,27,54]. In addition, the *A. oryze kexB* deletion has been reported to cause hyper-branching similar to *A. niger*, and increased amylase and protease production [24]. In *A. fumigatus*, a *kexB* deletion strain showed increased sensitivity towards CFW and CR, similar to our observations for the *ΔkexB* strain here (Figure 3). The sensitivity towards these compounds combined with the genome-wide expression profile confirm activation of the CWI pathway in *ΔkexB* strains. 

The observed changes in cell wall chitin content in the *A. niger ΔkexB* strain are in congruence with *A. oryzae*, where the deletion of *kexB* was shown to increase cell wall chitin along with a reduction in α-glucan. This reduction in cell wall α-glucan was suggested to be caused by a lack of KexB-processed α-glucan synthases [26]. In this study, we did not analyze cell wall content for levels of α-glucan; however, we did observe a very strong positive transcriptional response for both *agsA* (5.52 FC) and *agsC* (26.48 FC) in the *ΔkexB* strain. Similar to upregulation of chitin-modifying enzymes, we found increased transcriptional expression of *agnD*, an alpha-1,3-glucanase [43] and *agtB*, an α-glucan glycanosyl transferase [55] in conjunction with the two α-glucan synthases. Upregulation of α-glucan metabolism may either indicate an attempted compensatory response to a lack of correctly processed α-glucan synthases [56], or be a sign of general cell wall stress resulting from other improperly processed (cell wall) enzymes. Previously, we showed the relevance of proper α-glucan deposition in the cell wall for integrity; when α-glucan synthases *agsA* and *agsE* are disrupted in *A. niger,* we showed that *crh-*facilitated chitin-β-glucan crosslinking genes become important for maintenance of cell wall integrity, while these genes are redundant when α-glucan synthesis is undisturbed [31]. Interestingly, α-glucan has also been described in relation to pathogenesis and hyphal aggregation for filamentous fungi [56], by showing conidial and hyphal pellet clumping in α-glucan synthase knockouts of *A. fumigatus*, *A. oryzae* and *A. nidulans* [57,58,59]. Hyphal aggregation of the *A. niger ΔkexB* strain appeared to be reduced across all three pH conditions (Figure 1) which may be related to cell wall α-glucan levels. Further analysis of α-glucan-dependent hyphal aggregation and biofilm formation may be a topic of future research.

Next to upregulation of α-glucan and chitin metabolism, *fksA*, the sole β-1,3-glucan synthase in *A. niger* was also found to be upregulated. Along with *fksA*, differential expressions of glucanases and glucanosyltransferases (Table 2) were observed, that are predicted to cause rearrangements of β-1,3-glucan polymer length and branching. This transcriptional response in *A. niger* is very similar to that observed in the *kexB* deletion in *A. fumigatus,* that also resulted in upregulation of *gel* glucanosyltransferases [54], and the results described on the β-1,3-glucan composition in *A. oryzae,* where the *AoΔkexB* strain showed a higher degree of β-glucan polymerization combined with slightly less branching [26]. Together with the overall cell wall biosynthetic gene response, a deletion of *kexB* appears to bring about major cell wall content and compositional changes for all major constituents of the cell wall.

This study shows that the cell wall integrity of the *ΔkexB* strain is clearly affected, however, it is affected without impairing the growth rate or the maximally attained biomass during submerged batch cultivations. Contrarily, we did observe that *kexB* is important in the formation of biofilms, as this mode of cultivation in biofilm reactors led to a reduction in the ability to colonize the metal support. For now, understanding the characteristic phenotype of the *ΔkexB* strain during growth at pH 5.5 and 6 seems a consequence of pleiotropic effects of several hitherto unrelated responses. The morphology at pH 5.5/6.0 is possibly the result of loss of polarity due to the inability to synthesize or modify the cell wall correctly. Many possible cell wall synthesis or cell-wall-modifying enzymes could be targets of KexB processing. Explanations for the pH-dependent effect have been postulated and they suggest that at a lower pH, other proteases can perform KexB-like dibasic processing [28]. In yeast, yapsins, aspartic proteolytic enzymes involved in processing cell wall proteins including Scw4p, Utr2p, Pir4p and Gas1p [60,61,62], show a pH-dependent dependency on Kex2. For Scw4p, it was shown that only at a neutral pH (7.0), but not at an acidic pH (4.0), Kex2p is required for correct processing of the cell wall protein prior to yapsin processing, in order to activate the protein [62]. It is likely that the pH-dependent morphology of the ∆*kexB* mutant in *A. niger* is caused by improper processing of such enzymes/proteins. Here, we showed that—despite the clear morphological changes in liquid between pH 5.0 and 6.0—the respective increase in cell wall chitin compared to wild type was observed across all pH conditions. From this, we infer that the increase in chitin content is unlikely to dictate cell shape, however, we cannot rule out chitin-modifying enzymes could still play a role in the distinct phenotype. Taken together, the transcriptional data suggest activation of the CWI pathway that results in cell wall compositional rearrangements. Despite these findings, it remains to be elucidated how and if the KexB-dependent cell wall modifications are related to the pH-dependent growth phenotype.

## Figures and Tables

**Figure 1 microorganisms-08-01918-f001:**
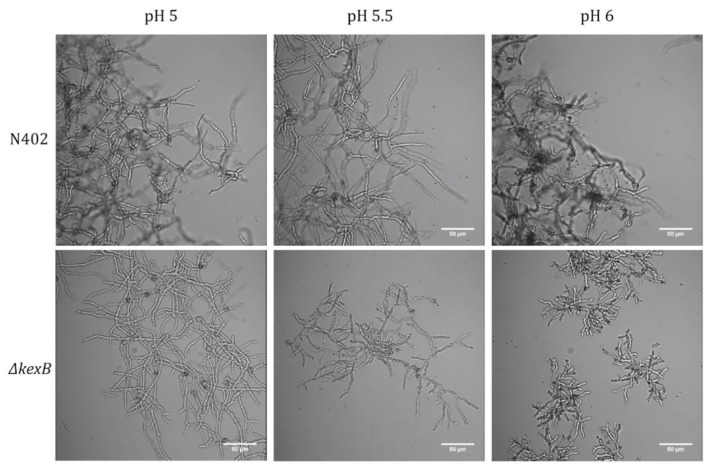
Pellet morphology of pH-controlled batch cultivations wild type (N402) and *∆kexB* strain microscopy samples taken at maximum biomass under different pH conditions: pH 5.0, pH 5.5 and pH 6.0.

**Figure 2 microorganisms-08-01918-f002:**
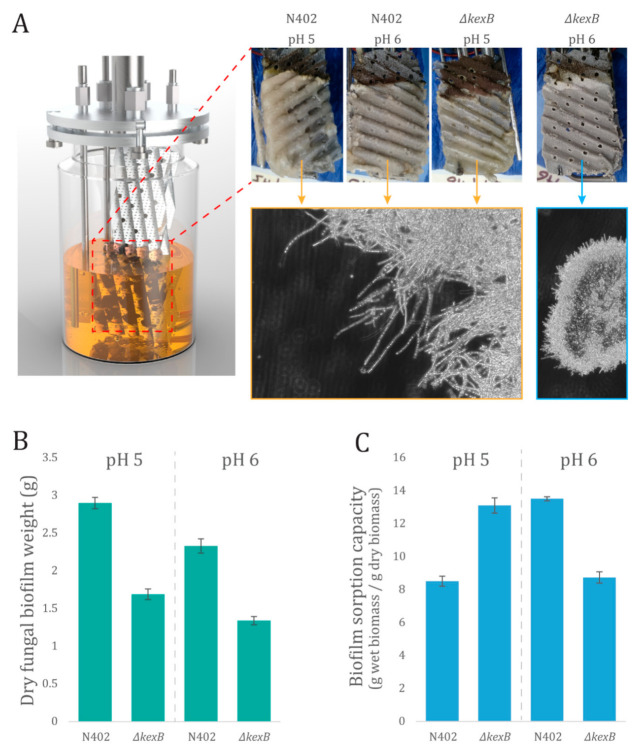
Submerged biofilm reactor cultivations of wild type (N402) and *∆kexB*. (**A**) Scheme of the biofilm reactor set-up used in this work and pictures displaying biofilm layer for each strain at two pH levels. Representative microscopy pictures (brightfield, 20×) are displayed for each type of biofilm structure. (**B**) Dry fungal biomass weight developed on the stainless-steel sheets in biofilm reactor and (**C**) biofilm water sorption capacity (*n* = 2). Biofilm (water) sorption capacity is the difference between biofilm’s wet and dry weight.

**Figure 3 microorganisms-08-01918-f003:**
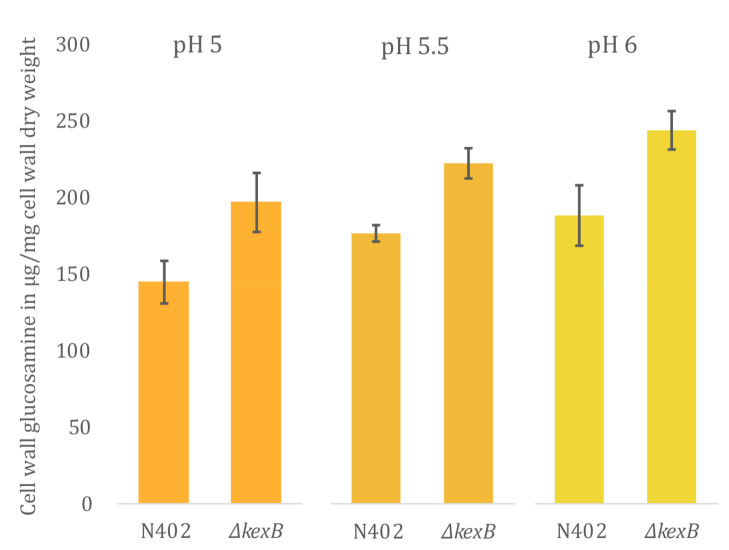
Cell wall glucosamine content of wild type (N402) and *∆kexB* from pH-controlled batch cultivations. Cell wall glucosamine content maximum biomass samples that originate from single bioreactor cultivations, run at either pH 5.0, pH 5.5 (1) or pH 6.0. Measurements were performed in technical triplicates (*n* = 3); error bars are the standard error (SE) and represent the variation of technical replicates.

**Figure 4 microorganisms-08-01918-f004:**
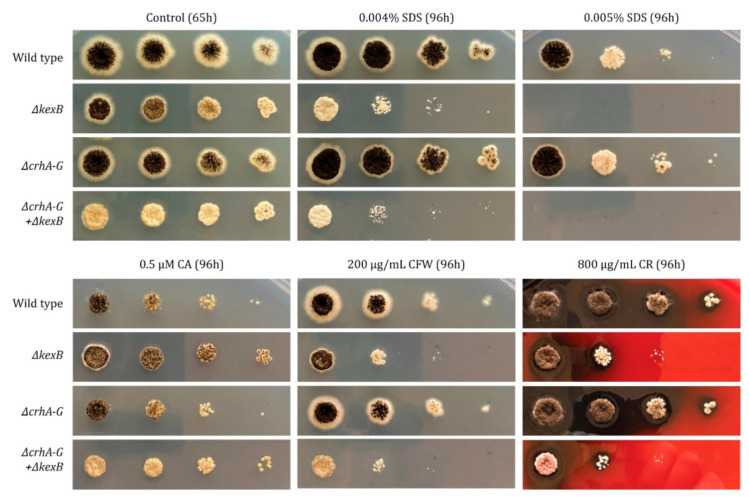
Susceptibility against cell wall disturbing compounds of wild type, *ΔkexB*, *ΔcrhA-G* and *ΔcrhA-G* + *ΔkexB*. Spore dilution assay on a range of concentrations of SDS, Caspofungin (CA), CalcoFluor White (CFW) and Congo Red (CR). The number of spores plated, from left to right, are 10^4^, 10^3^, 10^2^ and 10^1^ spores, respectively. All growth was performed in minimal medium (MM), pH 6.0. Strains were allowed to grow at 30 °C for 96 h prior to recordings, with the exception of the control (65 h).

**Table 1 microorganisms-08-01918-t001:** All strains used in this study.

Name	Genotype	Reference
N402	*cspA1*	[30]B
*∆kexB*	*cspA1, pyrG-, ΔkexB::AOpyrG (AB4.1∆pclA)*	[12]
TLF39	*cspA1, ΔkusA::DR-amdS-DR, ΔcrhA-G*	[31]
TLF69	*cspA1, ΔkusA::DR-amdS-DR, ΔcrhA-G, ΔkexB::hygB*	This study

**Table 2 microorganisms-08-01918-t002:** Growth parameters of wild type and *ΔkexB*. Growth is shown as maximum growth rate per hour (*µ*max) and maximum biomass (gDW·kg^−1^) at different pH-controlled batch-cultivations.

	Wild Type		*ΔkexB*	
pH Condition	*µ*max (h^−1^)	Max Biomass (g_DW_·kg^−1^)	*µ*max (h^−1^)	Max Biomass (g_DW_·kg^−1^)
pH 5.0	0.187	3.66	0.200	4.15
pH 5.5 (1)	0.175	3.68	0.194	4.19
pH 5.5 (2)	0.167	3.54	0.180	4.30
pH 6.0	0.063 *	0.64 *	0.211	4.31

* Cultivation of N402 at pH 6.0 resulted in severe biofilm formation on the walls of the fermenters resulting in underestimation of both the submerged growth rate and amount of in-broth maximum biomass.

**Table 3 microorganisms-08-01918-t003:** Differentially expressed cell wall biosynthesis genes in *A. niger*. 29 genes were differentially expressed (Padj ≤ 0.05) based on all 82 cell wall biosynthesis genes described by Pel et al., 2007 [41].

An ID	NRRL3 ID	Gene Description	Gene	Wild Type (Normalized Read Counts)	*ΔkexB* (Normalized Read Counts)	FC	Up- or Down-Regulated	Padj
**A-glucan biosynthesis and modification**								
An12g02460	NRRL3_09001	Putative GPI-anchored amylase-like protein (GH13-family) with possible function in alpha1,3-1,4-glucan processing	*agtB*	30	1114	38.06	up	1.43 × 10^−69^
An12g02450	NRRL3_09002	Putative catalytic subunit alpha1,3-glucan synthase complex; SpAgs1-like	*agsC*	64	1688	26.48	up	5.91 × 10^−67^
An04g09890	NRRL3_07454	Putative catalytic subunit alpha1,3-glucan synthase complex; SpAgs1-like	*agsA **	104	574	5.52	up	8.90 × 10^−25^
An08g09610	NRRL3_11494	Putative alpha-1,3-glucanase GH71; member of the SpAgn1-family	*agnD*	1496	6544	4.37	up	1.55 × 10^−71^
**Β-glucan biosynthesis and modification**								
An09g00670	NRRL3_00054	Predicted GPI-anchored protein. Putative 1,3-β-glucanosyltransferase GH72; member of the Gel-family	*gelD*	40	920	22.95	up	3.83 × 10^−73^
An03g05290	NRRL3_08399	Predicted GPI-anchored protein. Putative beta-1,3-glucanosyltransferase GH17; member of the AfBgt1-family	*bgtB*	20,956	29,235	1.40	up	7.26 × 10^−4^
An01g12450	NRRL3_02657	Putative exo-beta-1,3-glucanase (GH55-family); related to *Coniothyrium minitans* exo-1,3-glucanase (Cmg1)	*bxgA **	4743	6407	1.35	up	8.95 × 10^−3^
An06g01550	NRRL3_11624	Putative catalytic subunit beta1,3-glucan synthase complex; ScFks1-like	*fksA*	18,189	23,402	1.29	up	6.29 × 10^−4^
An07g04650	NRRL3_04586	Putative beta-1,3-glucanosyltransferase GH17; member of the AfBgt1-family	*bgtC*	1161	847	−1.37	down	2.46 × 10^−3^
An10g00400	NRRL3_06317	Predicted GPI-anchored protein. Putative 1,3-β-glucanosyltransferase GH72*; A. fumigatus* Gel1-like	*gelA **	14,085	9570	−1.47	down	4.10 × 10^−8^
An03g06220	NRRL3_08332	Predicted GPI-anchored protein. Putative 1,3-β-glucanosyltransferase GH72; member of the Gel-family	*gelE*	2008	192	−10.50	down	6.04 × 10^−120^
An19g00090	NRRL3_01223	Putative exo-beta-1,3-glucanase (GH55-family); related to *Coniothyrium minitans* exo-1,3-glucanase (Cmg1)	*bgxC*	1101	102	−10.83	down	3.28 × 10^−50^
**Chitin biosynthesis and modification**								
N/A	NRRL3_09653	Putative chitinase (N402 specific)	-	17	336	19.52	up	2.11 × 10^−17^
An12g10380	NRRL3_02932	Putative chitin synthase ClassIII; EnChsB-like	*chsE **	3764	6517	1.73	up	1.35 × 10^−15^
An08g05290	NRRL3_11152	Putative chitin synthase ClassVI;	*chsG*	147	252	1.71	up	4.56 × 10^−3^
An14g00650 An14g00660	NRRL3_00641	Putative chitin synthase ClassI; EnChsC-like	*chsC*	2398	3814	1.59	up	2.44 × 10^−7^
An11g01540	NRRL3_10021	Putative transglycosidase of GH16-family involved in cell wall biosynthesis; ScCrh1-like	*crhA*	737	1097	1.49	up	8.45 × 10^−4^
An01g11010	NRRL3_02532	Predicted GPI-anchored protein. Putative transglycosidase of GH16-family involved in cell wall biosynthesis; member of the ScCrh1-family	*crhD **	3637	5234	1.44	up	7.65 × 10^−4^
An09g02290	NRRL3_00179	Putative chitin synthase ClassIV; EnChsD-like	*chsD*	2278	3104	1.36	up	1.99 × 10^−3^
An09g04010	NRRL3_00331	Putative chitin synthase ClassIII; EnChsB-like	*chsB **	6774	8553	1.26	up	5.98 × 10^−3^
An01g05360	NRRL3_02063	Putative ClassV Chitinase (GH18); ScCts2-like	*cfcD*	1384	818	−1.69	down	3.96 × 10^−5^
An16g02850	NRRL3_07085	Putative transglycosidase of GH16-family involved in cell wall biosynthesis; member of the ScCrh1-family	*crhF*	1468	812	−1.81	down	4.92 × 10^−8^
An07g07530	NRRL3_04809	Predicted GPI-anchored protein. Putative transglycosidase of GH16-family involved in cell wall biosynthesis; ScCrh2-like	*crhB*	2205	988	−2.23	down	1.53 × 10^−9^
N/A	NRRL3_04221	Putative chitinase (N402 specific)	-	174	58	−2.99		3.09 × 10^−3^
An19g00100	NRRL3_01224	Putative ClassV Chitinase (GH18); ScCts2-like	*cfcG*	415	2	−197.88		8.05 × 10^−20^
**GH76 family proteins**								
An11g01240	NRRL3_10041	Putative endo-mannanase (GH76-family) with a possible role in GPI-CWP incorporation; ScDfg5-like	*dfgH*	289	1103	3.82	up	2.58 × 10^−35^
An14g03520	NRRL3_00897	Predicted GPI-anchored protein. Putative endo-mannanase (GH76-family) with a possible role in GPI-CWP incorporation; ScDfg5-like	*dfgC **	1133	1996	1.76	up	6.95 × 10^−10^
An16g08090	NRRL3_06700	Predicted GPI-anchored protein. Putative endo-mannanase (GH76-family) with a possible role in GPI-CWP incorporation; ScDfg5-like	*dfgE*	886	1169	1.32	up	2.14 × 10^−2^
An02g02660	NRRL3_06048	Putative endo-mannanase (GH76-family) with a possible role in GPI-CWP incorporation; ScDfg5-like	*dfgG*	2587	1693	−1.53	down	3.75 × 10^−5^
**Rho-GAPs**								
An18g06730	NRRL3_10703	Putative Cdc42-GTPase Activating protein (GAP) with similarity to ScBem3p	*capB*	1379	1783	1.29	up	5.54 × 10^−3^
An13g00850	NRRL3_01500	Putative Rho1-GTPase Activating protein (GAP) with strong similarity to ScRgd2	*rapE*	2256	2755	1.22	up	1.91 × 10^−2^
**CWI signaling**								
An04g10140	NRRL3_07436	Putative plasma membrane sensor required for cell wall integrity signaling; ScMtl1like	*mtlB*	93	575	6.16	up	6.51 × 10^−22^
An07g04070	NRRL3_04545	Putative plasma membrane sensor-transducer of the stress-activated PKC1-MPK1 kinase pathway involved in maintenance of cell wall integrity; ScWsc1-like	*wscB **	2073	2963	1.43	up	4.60 × 10^−5^
An18g02400	NRRL3_10351	Protein kinase C with putative function in CWI signaling	*pkcA*	3602	4518	1.25	up	5.43 × 10^−3^
An08g10670	NRRL3_11584	MAPK with putative function in Pheromone response/pseudohyphal growth pathway; ScFus3-like	*fusC*	2269	2808	1.24	up	1.39 × 10^−2^

* Identified as induced upon cell wall stress [42,43].

## Data Availability

The DNA reads described in this study are deposited in the GEO accession database under accession number GSE151618. All other data are available on request by contacting the corresponding author.

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
