# Peer review of "Deletion of the Aspergillus niger Pro-Protein Processing Protease Gene kexB Results in a pH-Dependent Morphological Transition during Submerged Cultivations and Increases Cell Wall Chitin Content"

_microorganisms, 2020, doi:10.3390/microorganisms8121918_

Round 1

Reviewer 1 Report

It seems that current version of the manuscript has investigated on the impact of deleting kexB in A. niger on the cell wall composition with respect to its shorter and thicker hyphae. Overall it is quite interesting paper.

Author Response

Find the attachment. 

Reviewer 2 Report

The work entitled "Deletion of the Aspergillus niger pro-protein processing protease gene kexB results in a pH-dependent morphological transition during submerged cultivations and increases cell wall chitin content" highlights the role that the kexB gene plays in the hyper-branching of the mycelium and the cell wall in the A. niger fungus in submerged culture and the relationship with the pH of the medium. Likewise, the analysis of ΔkexB cell walls under the tested pH conditions showed an increase in chitin content.

Given that the work provides important information to identify suitable production strains that can be used for cell wall harvesting after fermentation and given the growing interest in the use of post-fermentation mycelial debris to obtain chitin from the cell wall, I consider it suitable for publication in the MDPI Journal Microorganism. The work is well written, the experimental development is adequate, and it is well described. The authors have used an adequate methodology. The references presented are adequate. I consider that the paper fulfils the requisites to be published in Microorganism and therefore I recommend its publication without any changes.
